# Macro- and Microplastics in the Antarctic Environment: Ongoing Assessment and Perspectives

Emilia Rota [1,*], Elisa Bergami [2], Ilaria Corsi [1] and Roberto Bargagli [1]

1 Department of Physics, Earth and Environmental Sciences, University of Siena, Via P.A. Mattioli 4, IT-53100 Siena, Italy; ilaria.corsi@unisi.it (I.C.); roberto.bargagli@unisi.it (R.B.)
2 Department of Life Sciences, University of Modena and Reggio Emilia, Via G. Campi 103, IT-41125 Modena, Italy; ebergami@unimore.it
* Correspondence: rota@unisi.it

**Abstract:** The number of scientists and tourists visiting Antarctica is on the rise and, despite the management framework for environmental protection, some coastal areas, particularly in the Antarctic Peninsula region, are affected by plastic contamination. The few data available on the occurrence of microplastics (<5 mm) are difficult to compare, due to the different methodologies used in monitoring studies. However, indications are emerging to guide future research and to implement environmental protocols. In the surface and subsurface waters of the Southern Ocean, plastic debris >300 μm appears to be scarce and far less abundant than paint chips released from research vessels. Yet, near some coastal scientific stations, the fragmentation and degradation of larger plastic items, as well as microbeads and microfibers released into wastewater from personal care products and laundry, could potentially affect marine organisms. Some studies indicate that, through long-range atmospheric transport, plastic fibers produced on other continents can be deposited in Antarctica. Drifting plastic debris can also cross the Polar Front, with the potential to carry alien fouling organisms into the Southern Ocean. Sea ice dynamics appear to favor the uptake of microplastics by ice algae and Antarctic krill, the key species in the Antarctic marine food web. *Euphausia superba* apparently has the ability to fragment and expel ingested plastic particles at the nanoscale. However, most Antarctic organisms are endemic species, with unique ecophysiological adaptations to extreme environmental conditions and are likely highly sensitive to cumulative stresses caused by climate change, microplastics and other anthropogenic disturbances. Although there is limited evidence to date that micro- and nanoplastics have direct biological effects, our review aims at raising awareness of the problem and, in order to assess the real potential impact of microplastics in Antarctica, underlines the urgency to fill the methodological gaps for their detection in all environmental matrices, and to equip scientific stations and ships with adequate wastewater treatment plants to reduce the release of microfibers.

**Keywords:** Antarctica; plastics; occurrence; environmental contamination; impact; marine and terrestrial ecosystems; ASPAs

## 1. Introduction

Plastics are a chemically diverse group of polymeric-based materials, obtained through different physical (e.g., melting, extrusion, pelletization) and chemical processes (e.g., mixing with plasticizers, colorants, copolymers and other compounds). For their durability, malleability and the low cost, plastic objects are globally used in a wide range of applications and in every-day life. Their production increased exponentially during the last 70 years and the current global production (>350 million tonnes year$^{-1}$) exceeds that of all the other manufactured chemicals [1]. More than 90% of plastics are made from fossil fuels and their production and disposal are responsible for the emission of a high carbon load. Moreover, a large proportion of plastic is used for packaging and, due to its very short life cycle, contributes a significant amount of domestic and industrial solid wastes. Since

plastics are of different types and often composite materials, their recycling is complicated and only a small percentage of plastic waste generated to date is recycled or incinerated; the rest is dumped in landfills [2–4]. In waters and soils, plastic litter undergoes degradation processes by ultraviolet light, waves, heat and microorganisms, and can form tiny 'secondary' fragments (generally classified as microplastics if 1–5000 µm in size; nanoplastics if <1 µm), which move in the environment [5–7]. Many other particles primarily having micro- to nanometer sizes, such as micro- and nanobeads in personal care products, or polypropylene, polyethylene, polyester, polyamide and acrylic microfibers from clothing, disposable diapers or cigarette filters, are released directly into the environment. Unlike natural textile fibers such as wool, cotton, silk or semisynthetic rayon, plastic microfibers are transformed and degraded very slowly. Depending on the polymer composition and environmental circumstances, micro- and nanoplastics (hereafter collectively referred to as MPs) can remain in the environment for decades to centuries and make up one of the most pervasive and ubiquitous synthetic materials in marine and terrestrial ecosystems worldwide [8–10]. Through weathering processes, they change their physicochemical properties with the release of dyes, plasticizer additives and other contaminants, and the formation of functionalized oxygen moieties on their surfaces [11]. Phthalates leached from polyvinyl chloride, for instance, are hydrophobic with an affinity for lipids and tend to get adsorbed on organic matter and accumulate in living organisms [12]. Oxidative stress is a rather common response of organisms exposed to MPs; furthermore, ingested plastic particles can block the digestive system of animals and can also act as vectors for environmental pollutants and microorganisms [13,14]. In recent years, the increasing evidence of direct or indirect biological effects of plastic litter and thus, the awareness that these pollutants are a relevant issue for the conservation of biodiversity, have promoted an extraordinary increase in research on environmental MPs on a global scale [15–17].

Antarctica is regarded as the last great wilderness on Earth. Human presence in the Antarctic Treaty Area is limited to scientific research and support for logistics, tourism and fishing, and an environmental impact assessment is required for most anthropogenic activities. The Protocol on Environmental Protection to the Antarctic Treaty (commonly known as the "Madrid Protocol", adopted in 1991 and entered into force in 1998) has designated Antarctica as "a natural reserve, devoted to peace and science", and provisions associated with Annex III (Waste Disposal and Waste Management) and Annex IV (Prevention of Marine Pollution) regulate all activities occurring south of 50° S latitude in the Atlantic, 45–55° S latitude in the Indian Ocean and 60° S latitude in the Pacific. Nevertheless, this remote region is not immune from the impact of local or remote human activities, especially in the Antarctic Peninsula region, where there has been a remarkable increase in scientific stations and tourism activities (more than 70,000 visitors in the 2019–2020 season) [18–20]. Despite the comprehensive management framework for environmental protection, the region is increasingly threatened by the impact of global warming, the introduction of alien and invasive species and by persistent contaminants coming from local and remote sources [21–24]. As in the rest of the world, also in the most visited Antarctic areas, macroscopic plastic debris often constitutes one of the most enduring and frequent evidence of past and recent human activity. The entanglement of fur seals in plastic waste as well as the occurrence of plastic particles in the stomach of petrels breeding on the continent have been reported since the 1980s [25,26]. In subsequent years, there have been several reports of MPs ingestion by marine predators, and plastic debris has also been detected in Antarctic Specially Protected Areas (ASPAs) [27–32]. The latter areas, by virtue of their outstanding environmental, scientific, historic or wilderness values, are accessible (with permission and post-visit reporting) only for scientific activities that do not pose a threat to the protected values.

The attention and concern for plastic pollution in Antarctica by the scientific community and by institutions included in the Antarctic Treaty System have only recently increased [33–39]. The last 5 years have seen a proliferation of studies on the presence of plastic debris in marine ecosystems (sea ice, waters, sediments and beaches); other studies

have focused on the terrestrial environment (soils and snow) and on the occurrence of MPs in Antarctic organisms and food webs [9,40–59]. Although several laboratory studies have found no MPs toxicity in exposed organisms and although concentrations of other persistent contaminants measured in Antarctic biota are lower than those causing biological effects on related species in temperate regions or the Arctic, it seems reasonable to worry about the potential impact of MPs in the Antarctic biota [21,60,61]. In fact, most Antarctic organisms are endemic species and have unique ecophysiological adaptations to extreme environmental conditions, resulting from long evolutionary histories in isolation [62–66]. Until recently they were subject to little anthropogenic influences and are probably among the organisms most vulnerable to cumulative stresses caused by contaminants and changes in climatic and environmental conditions.

Although the growing abundance of MPs in the global environment has catalyzed the interest of the scientific community, we still need to fill knowledge gaps related to their real potential risks and solve technical challenges such as standard methods for extraction, quantification and characterization of particles < 100 µm in different environmental media. These problems are even more difficult to address in Antarctica, where high UV levels in the austral summer, low temperatures and seasonal sea ice formation can affect thermal oxidation and the environmental life cycle of plastic debris. The entrapment of plastics during sea ice formation could promote their degradation, increasing their bioavailability for phyto- and zooplankton, especially during seasonal melting (Figure 1). Furthermore, in Antarctica MPs interact with biota and food webs unique on Earth, where organisms usually have slower development and generative turnover, and very narrow temperature tolerance. The growing impact of scientific and tourism activities, together with changes in local climate and environmental conditions, could produce a cascade of events that would affect the structure and functioning of marine and terrestrial ecosystems [18,19].

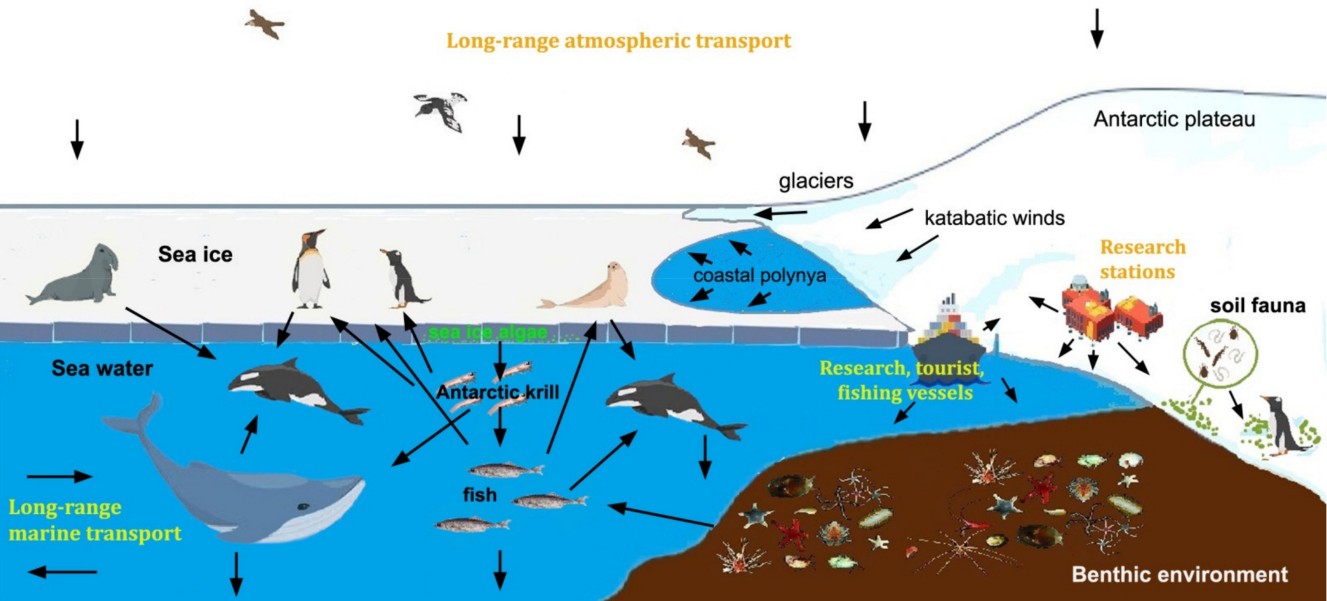

**Figure 1.** Sources, routes and fate of microplastics in the Antarctic environment.

This review will collect and discuss the available information on the sources, occurrence and distribution of MPs in Antarctica in the context of regional climatic and environmental conditions and the unique ecophysiological features of the Antarctic biota. A better knowledge of the occurrence and potential biological impact of MPs and other anthropogenic and climatic stressors is an indispensable prerequisite for implementing environmental protocols and management strategies for the Antarctic environment. Future research needs and requirements for routine standardized monitoring programs will also be highlighted.

## 2. Sources and Occurrence of MPs in the Southern Ocean

The opening of the Drake Passage between the Antarctic Peninsula and South America (about 30 Ma ago), gave rise to the Antarctic Circumpolar Current, resulting in gradual cooling and isolation of the continent and the formation of the Antarctic cyclonic vortex. Nevertheless, the local occurrence of persistent environmental contaminants neither produced nor used in Antarctica indicates their arrival with air and water masses from lower latitudes [18]. Apparently, the long-range atmospheric transport of MPs (mostly fibers) reported for other remote regions [67] can also affect the Antarctic environment. Similarly, although the Polar Front was believed to be an efficient barrier to the southward transfer of floating or submerged materials, more and more data, including surveys of satellite-tracked ocean drifters or floating kelps, indicate their southward transfer [68]. Thus, although the disposal of waste and wastewater from research, tourism and fishing vessels, as well as from research stations and refuge huts, remains the main (local) source of plastic contamination, there is mounting evidence that the Southern Ocean and Antarctica are receiving MPs from lower latitudes. The positive buoyancy and chemical stability of plastics probably allow them to cross the Polar Front through meandering and the generation of eddies. Furthermore, wind-driven turbulence and vertical mixing increase the concentrations of the smallest MPs in deep waters [69,70], and their upwelling in the Antarctic Circumpolar Current could favor southward transport. In the Southern Ocean, MPs can be dispersed by the gyres of the Weddell and Ross Seas and the Antarctic Coastal Current [38]. MPs can also leave the water through bubble burst ejection and spume, thus becoming part of air masses and being transported and/or entrapped in snow and sea ice [70]. In coastal polynyas, for instance, MPs are incorporated in newly formed sea ice, which is pushed by continental winds offshore [71] (Figure 1).

Available data on the geographical distribution and characteristics of macro- and microplastics in Antarctica are still scarce and patchy (Table 1). Most surveys have been carried out in the West Antarctic Peninsula and the Ross Sea regions; different methods have been used for the sampling, detection and identification of plastic polymers and, sometimes, only the concentrations of floating macro-litter (>25 mm) have been determined [5,72–75]. Further uncertainties arise from the heterogeneity in polymer types and the lack of knowledge regarding their degradation processes in the cold Antarctic environment. While many large fragments float on the sea surface and are exposed to the UV, wind and ice action, MPs are usually submerged, move more slowly and can be lost during the sampling (their size being smaller than the commonly used mesh size: >0.1 mm) [76,77].

**Table 1.** Microplastics in Antarctic environmental abiotic matrices.

| Matrix | Location | Depth | Collection Identification | Shape | Size | Concentration (Items) | Ref. |
|---|---|---|---|---|---|---|---|
| Snow | East Antarctica, Ross Island (McMurdo and Scott Base, up to 20 km away) | 0–2 cm | Manual μFTIR * | Fibers (61%), Fragments, Films | Mean: 606 μm Range: 50–3510 μm | Mean: $29.4 \pm 4.7\,\text{L}^{-1}$ (of melted snow) Range: $4–82\,\text{L}^{-1}$ | [48] |
| Freshwater | ASPA 126 Livingston Island (Byers Peninsula) | 0 m | Drifting nets (333 μm and 100 μm mesh) μFTIR * | Fibers, films | Mean: 1118 μm Range: 400–3546 μm | Range: $4.7–14.3 \times 10^{-4}\,\text{m}^{-3}$ (EPS) | [78] |
| Ice | King George Island (Bellingshausen Dome; "Collins Glacier") | ——— | Visual and manual FTIR ** and μFTIR * | Fragments | Range: 2.3–12.6 mm | Mean: $9.5 \times 10^{-4}\,\text{m}^{-3}$ Range: $0.17–0.33\,\text{m}^{-2}$ | [79] |

**Table 1.** *Cont.*

| Matrix | Location | Depth | Collection Identification | Shape | Size | Concentration (Items) | Ref. |
|---|---|---|---|---|---|---|---|
| Coastal landfast sea ice | East Antarctica (North Casey Station) | 0–115 cm | Stainless-steel corer Automated µFTIR * | ———— | Mean: 56.7 µm Range: 20–325 µm | Mean: 20.4 L$^{-1}$ (of melted ice) Range: 6–33 L$^{-1}$ | [41] |
| Coastal landfast sea ice | East Antarctica Ross Sea, Cape Evans | 0–150 cm | Stainless-steel corer TD-PTR-MS *** | ———— | <200 nm | Top of the core: 67 ng mL$^{-1}$ Bottom: 37.7 ng mL$^{-1}$ | [80] |
| Seawater | King George Island (Admiralty Bay) | 0–30 m | Neuston net (150 µm mesh) µRaman | Fibers | Range: ca. 2–5 mm | Mean: 0.024 ± 0.0457 m$^{-3}$ | [81] |
| Seawater | East Antarctica, Ross Sea (near-shore and offshore) | 5 m | Pumped water (each sample from 600 to 2000 m$^3$) µFTIR * | Fragments (72%), Fibers (13%) | >60 µm | Mean: 0.17 ± 0.34 m$^{-3}$ Range: 0.0032–1.18 m$^{-3}$ | [42] |
| Seawater | Mid Scotia Sea and Adelaide Island (55°S to 67°S) | Surface waters | Microplastic net (300 µm mesh) µFTIR * | Fragments (93%), Film (7%) | 90% <300 µm | Mean: 0.013 ± 0.005 m$^{-3}$ (5056 ± 2158 km$^{-2}$) Maximum: 0.054 m$^{-3}$ | [56] |
| Seawater | Weddell Sea | Surface waters Subsurface (11.2 m) | Manta net (300 µm mesh) FTIR ** Pumped and filtered (300 µm) FTIR ** | Fragments (90.2%), Lines (8.8%), Fragments (100%) | 74% <900 µm 64% <700 µm | Mean: 0.01 ± 0.01 m$^{-3}$ Range: 0–0.04 m$^{-3}$ Mean: 0.04 ± 0.1 m$^{-3}$ Range: 0–0.47 m$^{-3}$ | [82] |
| Seawater | Off the Antarctic Peninsula | Surface waters | Manta net (330 µm mesh) FTIR ** | Fragments (51.3%), "Line" (42.3%),"Sphere" (6.4%) | <5 mm (54%) >5–200 mm (46%) | Mean: 1794 km$^{-2}$ (0.008 items m$^{-3}$) Range: 755–3524 km$^{-2}$ | [83] |
| Seawater | Southern Ocean | Surface waters | Neuston net (350 µm mesh) FTIR ** | Fragments | <5 mm | Range: 0.03–0.09 m$^{-3}$ | [84] |
| Seawater | Southern Ocean | Surface waters | Neuston net (200 µm mesh) µFTIR * | Fragments | Mean: 3.03 mm Range: 0.68–21.5 mm | Mean: 188 km$^{-2}$ | [8] |
| Seawater | Southern Ocean | Surface and subsurface (5 m) waters | Stainless-steel bucket/Ship underway pump µFTIR * | Fibers (72.9% cellulose) | Median: approx. 0.9 mm | Median: 1.7 L$^{-1}$ $Q_1$–$Q_3$: 0.4–3.5 L$^{-1}$ | [85] |
| Sediment | East Antarctica, Ross Sea (Terra Nova Bay) | 25, 70 and 140 m | Van Veen grab FTIR ** | Fibers (42.8%), Film (35%), Fragments (22.2%) | 78.4% < 5 mm | Range: 5–1705 m$^{-2}$ | [45] |
| Sediment | Antarctic Peninsula Adelaide Island (Rothera Station) | 0–20 m | Diving or box coring FTIR ** | Fibers (nearly all) | <5 mm | Range: 0–3 mL$^{-1}$ | [47] |
| Sediment | King George Island (Admiralty Bay) | 6–60 m | Van Veen grab and SCUB FTIR ** | Fibers and Fragments | <5 mm | Range: 16–766 m$^{-2}$ | [38] |

**Table 1.** *Cont.*

| Matrix | Location | Depth | Collection Identification | Shape | Size | Concentration (Items) | Ref. |
|---|---|---|---|---|---|---|---|
| Sediment (cores) | Antarctic Peninsula, South Sandwich Islands, South Georgia Island | 136–3633 m | OKTOPUS Multicores Manual µFTIR * | Fragments (56%),Fibers (39%) | <2 mm | Mean: Antarctic Pen. $1.30 \pm 0.51$ g$^{-1}$, S Sandwich Is. $1.09 \pm 0.22$ g$^{-1}$, S Georgia Is. $1.04 \pm 0.39$ g$^{-1}$ | [86] |

* µFTIR: Fourier Transform InfraRed microscopy. ** FTIR: (micro-)Fourier Transform InfraRed spectroscopy. *** TD-PTR-MS: Thermal Desorption—Proton Transfer Reaction—Mass Spectrometry.

Using the data available on human presence in Antarctica in the period 2004–2014 and the amount of MPs produced daily per person through personal care products and laundry, Waller et al. [38] estimated for 2015 a release in the Southern Ocean of primary MPs ranging from 7.5 to 27.5 mg person$^{-1}$ day$^{-1}$. The impact of microplastics from coastal scientific stations, tourism and legal fishing was considered significant on a local scale, but likely negligible on the Southern Ocean scale. Although the complete removal of MPs is difficult even in plants with tertiary microfiltration treatments, these results suggest, where possible, equipping vessels and scientific stations with suitable wastewater treatment plants. Currently, over 30% of permanent bases and about 60% of those operating in the summer do not have one, and where they have been installed, they often show operating problems and malfunctions [87,88].

In any case, available data from surveys of macro- and microplastics on Antarctic beaches and throughout the Southern Ocean indicate that, unlike most other seas where about 75–90% of plastic fragments originate from land, coastal scientific stations have very localized impacts, and a large proportion of plastics in the Southern Ocean comes from fishing, tourism and research vessels and, probably, from long-range transport from lower latitudes [38,89]. In the ocean, plastic debris is carried and accumulates in the center of gyres. By filtering over 8 million liters of surface and subsurface waters from the Weddell Sea gyre, Leistenschneider et al. [82] found that 45.5% of all recovered fragments originated from paints of the research vessel; mean concentrations of plastic fragments (>300 µm) were $0.01 \pm 0.01$ m$^{-3}$ and $0.04 \pm 0.1$ m$^{-3}$, for surface and subsurface waters, respectively (Table 1). By comparing the results of two expeditions in the Weddell Sea, those authors highlighted a great temporal variability and found that no MPs occurred in water samples collected from coastal polynyas and east of the Antarctic Peninsula. The much greater occurrence of paint fragments in the Southern Ocean than in other seas was attributed by Leistenschneider et al. [82] to the low contamination by other plastic polymers and the increased mechanical abrasion of ship paints in the cold water of the Southern Ocean. In surface waters off the Antarctic Peninsula, Lacerda et al. [83] found paint fragments at all sampling stations and their number was 30 times that of plastics. Similarly, Jones-Williams et al. [56] reported that 45.6% of the microplastics collected in surface waters from the same region were paint chips originating from the ship. In a survey extending from temperate waters to the Southern Ocean, Suaria et al. [8] found that paint fragments from ships in neuston samples were 58% of all collected particles. Although this sub-group of microplastics has been overlooked in ecotoxicity studies, paint fragments could have deleterious effects on Antarctic marine organisms through direct ingestion or absorption of leached compounds, and persistent organic pollutants associated with paint have already been detected in phytoplankton samples collected off Deception Islands [90]. Due to their specific gravity and processes occurring in the marine environment, such as biofouling or attachment to marine snow, fecal pellets and plankton organisms, paint fragments settle and accumulate in sediments, where they can be ingested by the benthic fauna [91] with unknown impacts on the biota.

In subsurface (5 m, near-shore and offshore) water samples from the Ross Sea, Cincinelli et al. [42] found much more MPs (mean concentration $0.17 \pm 0.34$ particles m$^{-3}$) than those reported from the Weddell Sea ($0.04 \pm 0.1$ particles m$^{-3}$ [82]). Although comparisons are difficult, because smaller MPs, including fibers, were considered in the Ross Sea waters (Table 1), the polymer composition in the latter sea was different, probably due to the greater impact of sewage from coastal stations and from research and logistic vessels [42]. In general, the available data indicate that the waters of the Southern Ocean have some of the lowest concentrations of plastics globally. Many fragments are secondary MPs, originating from the weathering of larger pieces (often polyamide and polyurethane) rather than common polymers in single-use plastics, which are the most abundant in other seas. However, even in the most disturbed coastal ecosystems, it is impossible to estimate the potential impact of plastics. There is a lack of information on particles of few micrometers or sub-micrometer sizes (nanoplastics), which can also be transported by air masses. Materić et al. [80] have recently found higher concentrations of MPs in an Antarctic Sea ice core than in seawater. Through Thermal Desorption-Proton Transfer Reaction-Mass Spectrometry (TD-PTR-MS) analysis, they found that polyethylene and polypropylene were the most abundant particles and, unlike previous surveys on Antarctic Sea ice and surface waters [41,42], they also observed significant amounts of polyethylene terephthalate. In September, the extension of sea ice around Antarctica may exceed 18 million km$^2$, and too little is known regarding the behavior of MPs at the seawater/ice interface. A recent study, for example, suggests that MPs are selectively incorporated into the sea ice and nanoplastics are expulsed into the water below [92]. Therefore, nanoplastics could be more available to algae living on the underside of sea ice (Figure 1).

Surveys on macro- (>1 cm) and mesoplastics (5–10 mm) have been carried out mainly on beaches of the Antarctic Peninsula, sub-Antarctic and Maritime Antarctic islands [31,46,53,93]. Waluda and coworkers [94] surveyed beached marine debris and other anthropogenic waste materials (>5 mm) at two sites in the Scotia Sea (Southwest Atlantic sector of the Southern Ocean) for three decades. The total mass of the items collected was 101 kg, with an average accumulation rate of 100 items km$^{-1}$ month$^{-1}$. Plastic fragments were far more prevalent (97.5% by number and 89% by mass) than chips of glass, metals, paper or rubber. On Signy Island (the South Orkney Islands, within the Antarctic Treaty area), items were collected only during the austral summer (plastic debris 80% by mass) and the average accumulation rate was 3 items km$^{-1}$ month$^{-1}$. The few studies on MPs in marine sediments have focused mainly on samples collected near coastal scientific stations [45–47]; therefore, fibers from clothes washing were the most frequent types of MPs and their concentrations were in the same range as those recorded in other sediments collected outside Antarctica. However, MPs have been also detected in deep-sea sediments of the Southern Ocean. Cunningham et al. [86] analyzed 30 sediment cores from the Antarctic Peninsula, the South Sandwich Islands and South Georgia and found MPs in 93% of the samples, with a mean concentration >1 MP item g$^{-1}$ of sediment. The distribution and accumulation of MPs in the sedimentary environment reflected that of low-density particles, such as clay. Interestingly, unlike other MPs, microfibers did not show a preferential setting in low-energy environments and their distribution pattern suggested long-range atmospheric transport and widespread deposition throughout the Southern Ocean [86]. The marine areas investigated in this survey are among the most impacted by human activities in Antarctica, so MPs concentrations were found to be higher than those in other parts of the Southern Ocean and in the same range as those reported in other remote marine areas [95].

## 3. Microplastics in Marine Organisms and Food Webs

Ingested MPs can have direct negative effects on marine organisms and can act as a vector for other persistent organic pollutants, which can be biomagnified by trophic transfer in the top predators [13–15]. Recent studies suggest a possible multigenerational impact of MPs through the transfer of neurotoxicity, reproductive toxicity and oxidative stress from parents to offspring [96,97]. Due to their persistent and hydrophobic nature,

millimeter-sized plastic debris in the marine environment can stimulate the formation of surface biofilms and the establishment of "epi-plastic" communities. Colonized fragments can harbor viruses, bacteria, fungi, algae and marine invertebrates such as cnidarians, bryozoans or barnacles [98]. Pathogenic microorganisms have also been detected in epi-plastic communities [99], and Arias-Andres et al. [100] found that epi-plastic bacteria have a higher gene transfer frequency than free-living bacteria. This increased gene exchange occurs in a broad range of phylogenetically-diverse bacteria and could cause the spread of antibiotic resistance, affecting aquatic microbial communities on a global scale and posing a potential hazard to human health. On plastic and paint fragments collected around the Antarctic Peninsula, Lacerda et al. [83] found coccoid and filamentous bacteria, microalgae and invertebrates. The analysis of prokaryotic communities in two plastic fragments from King George Island showed the dominance of Gamma- and Betaproteobacteria in the one taken from the sea and Beta- and Alphaproteobacteria in the one collected on land [53]. In a piece of polystyrene stranded on the shore of the same island, Laganà et al. [101] isolated bacterial strains with multiple antibiotic resistance. Although in most waters and sediments of the Southern Ocean the concentrations of macro- and microplastics are much lower than in other seas, paint and plastic debris can be perceived as food items by marine consumers; furthermore, long-range transport of epi-plastic organisms could introduce alien species into Antarctic fouling communities.

The entanglement of marine mammals and birds is the most documented impact of plastics on Antarctic organisms. Do Sul et al. [73], reviewing literature data from 1982 to 2010, reported that more than a thousand fur seals (*Arctocephalus gazella*) were entangled at South Georgia and South Orkney islands, mainly with neck collars made of nylon ropes, packaging plastic bands, rubber rings or other fishing items. The wandering albatross (*Diomedea exulans*) and other species of marine birds were among the most affected by fishing gear and plastic debris. Accidental mortality (bycatch) in longline fishing for the Patagonian toothfish (*Dissostichus eleginoides*) has been estimated to be about 6000 seabirds around South Georgia Island and 13,000–14,000 around Crozet and Kerguelen Islands each year [102]. Since the late 18th century, a few million whales, fur seals and southern elephant seals have been killed in the Antarctic Treaty Area. However, in 1978 the Convention for the Conservation of Antarctic Seals came into force, and in 1994 the Southern Ocean was designated as a whale sanctuary by the International Whaling Commission. Thus, most historically harvested populations are recovering and the Commission for the Conservation of Antarctic Marine Resources (CCMLAR), which manages fisheries in the Southern Ocean, is implementing mitigation measures to reduce the entanglement of seabirds and marine mammals [102]. However, these animals are not immune from the ingestion of plastic debris. Philips and Waluda [103], for instance, analyzed macro- and mesoplastics (excluding fishing gears) associated with seabirds at South Georgia Island for 26 years and found a long-term increase in ingested debris for two species of albatrosses. It must be said that most of the plastic items originated from South America and, as confirmed by data from Antarctic prion [104] and penguins [55,105], MPs contamination is much lower than in seabirds from other regions worldwide. Fragão et al. [106] collected scat samples from breeding colonies of three species of penguins in the Antarctic Peninsula and the Scotia Arc over seven seasons. They found MPs in 15%, 28% and 29% of the Adélie, chinstrap and gentoo penguin scat samples, respectively, but no clear temporal variations or differences in the occurrence of particles across the colonies. Seals have also been used as biomonitors of plastic contamination. Erikson and Burton [107] extracted 164 plastic particles (mean length 4.1 mm) from 145 fur seals scats on Macquarie Island; most particles originated from the breakdown of larger fragments and acquired their final form by active abrasion. The authors hypothesized that seals accumulated plastics mainly through their food (i.e., the pelagic fish *Electrona subaspera*). Ryan et al. [108] found small plastic fragments (mainly 2–5 mm in size) in scats of fur seals breeding on Macquarie Island; however, they did not detect MPs in samples of seal scat from other sub-Antarctic islands. South of the Antarctic

Polar Front, contamination levels must be very low, as no MPs have recently been found in 42 male fur seals scats from Deception Island [109].

In the Arctic, von Friesen and coworkers [110] demonstrated that the release of MPs during the summer melting of sea ice was concomitant with the ice-edge bloom of sympagic and pelagic marine communities. In East Antarctic sea ice, Kelly et al. [41] found a positive relationship between concentrations of MPs and those of chlorophyll *a*, suggesting that ice algae can contribute to the entry of MPs into food webs. Moreover, the Antarctic krill *Euphausia superba* is a fundamental food source for many consumers, and sea ice diatoms are essential for the overwintering of the young krill (age class 0) [111]. Fragments of polyethylene, polypropylene and expanded polystyrene are less dense than seawater and are available for the ingestion and egestion by planktonic species. The Antarctic krill exposed to polyethylene microspheres for 10 days showed high uptake and depuration rates without bioaccumulation and acute toxicity [112]. In another exposure experiment, ingested plastic particles (31.5 μm) were partly fragmented by *E. superba* into pieces small enough (less than 1 μm) to cross physical barriers and potentially interact at the molecular level. Probably, MPs fragmentation is enhanced by the presence of silica diatoms in the diet of krill; anyhow, this study indicates that this keystone Antarctic species, through the production of nanoplastics, can affect the biogeochemical cycle and environmental fate of plastic contaminants in the Southern Ocean [113]. Bergami et al. [58] investigated the acute short-term toxicity of plastic nanospheres (50–60 nm) on *E. superba* juveniles and noted impairments in moulting and swimming and the excretion of nanoplastics via fecal pellets, suggesting potential effects on the biological carbon pump in the Southern Ocean.

Global trends of ocean acidification are exacerbated in the Southern Ocean by the low concentrations of carbonate ions and low water temperatures ($-1.8$ °C), which enhance the solubility of $CO_2$ [114]. Although most studies on the impact of ocean acidification have focused on calcifying organisms, there is evidence that decreasing pH values can also affect physiological functions in non-calcifying marine invertebrates. Regarding Antarctic krill, Kawaguchi et al. [115] found inhibition of embryonic development in *E. superba* at simulated 2000 μatm ρ$CO_2$, whereas, more recently, Ericson et al. [116] showed that adult krill is resilient to exposure for one year to near-future levels of ocean acidification (1000–2000 μatm ρ$CO_2$). However, possible effects cannot be ruled out for larval Antarctic krill, which throughout furcilia stages lives in close association with sea ice (i.e., an important plastic sink) and is concomitantly exposed to climate stressors, such as ocean warming and acidification. Rowland et al. [117] investigated the combined effects of different nanoplastics and ocean acidification on the embryonic development of Antarctic krill and observed a lower proportion of developing embryos in the multi-stressor treatment. As calcifying organisms, pteropods are certainly exposed to ocean acidification. Manno et al. [118] evaluated the combined effects of nanoplastics and low pH (7.8) on the sub-Antarctic pteropod *Limacina retroversa*, and even a short-term exposure (48 h) had negative effects on its survival. These studies suggest that ecotoxicological experiments aiming at assessing the impact of micro- and nanoplastics and other anthropogenic contaminants should consider additional stressors, which can alter the sensitivity thresholds of the organisms tested.

Most Antarctic marine species live on the continental shelf, but very little is known regarding the impact of MPs on benthic communities. Sfriso et al. [52] analyzed 12 macrobenthic species with different feeding strategies from three coastal sites at an increasing distance from the sewage treatment plant outfall at the "Mario Zucchelli" Italian Scientific Station. A large proportion (83%) of the species analyzed, and especially those collected near the station contained plastic particles (0.01–3.29 items mg$^{-1}$, mostly 50 to 100 μm in size). In agreement with the survey on MPs in surface sediments of the same marine area [45], the scientific station turned out to be the main source of MPs. Interestingly, the much higher accumulation in filter-feeding organisms than in predators and omnivores suggested that there was probably no MP bioaccumulation along the benthic food web [52].

### 4. Plastic Debris in Snow, Ice and Terrestrial Ecosystems

The increasing human footprint is reducing the pristine nature of Antarctica and its value as a reference area for the study of global processes. Continental ice contains most of the world's freshwater and is a key source for insights into Earth's history such as paleoclimate, meteorite impacts or the occurrence of extremophilic organisms in subglacial lakes. However, in freshwater glaciers within an ASPA at King George Island (used for long-term ecological monitoring and as a reference for inland water research), González-Pleiter et al. [79] found a ubiquitous occurrence (0.17 to 0.33 items m$^{-2}$) of expanded polystyrene particles, probably transported by wind from local sources. Aves et al. [48] analyzed snow samples collected near McMurdo and Scott Base research stations and from sites up to 20 km away. They found MPs (mainly fibers) in all samples (average concentrations 29 particles L$^{-1}$) and, retracing the trajectories of air masses from the time of sampling, they identified MPs from local sources (mainly polymers used in clothing and equipment in scientific stations) and from potential long-range transportation. By assuming for the fibers an atmospheric residence time of 6.5 days, it was estimated that potential sources of plastic fibers were located at a distance of up to 6000 km [48]. The atmospheric contribution of MPs to Antarctica is unknown, however, this estimate is not surprising. The calculated global atmospheric emissions of plastic microparticles are about 9.6 ± 3.6 Tg year$^{-1}$ and those of microfibers are 6.5 ± 2.9 Tg year$^{-1}$ [119]. Due to their small size and low density, MPs are transported very efficiently by winds around the Earth [120] and, like other persistent organic pollutants, can undergo grasshopping processes (i.e., a number of depositions followed by resuspension). In the atmosphere, MPs can influence the Earth's climate by acting as ice nuclei in clouds and by absorbing and scattering solar radiation (estimated radiative forcing = 0.044 ± 0.399 mW m$^{-2}$); moreover, when deposited on snow and ice, they may accelerate the melting of the cryosphere [121].

Most scientific stations and human activities in Antarctica are concentrated in ice-free coastal areas and although there is evidence that sites with high anthropogenic influence are impacted by macro- and MPs pollution, terrestrial systems have received far less attention than marine systems. Between November 2019 and January 2020, Finger et al. [49] recovered 1544 anthropogenic items within the ASPA No. 133 at Harmony Point, Nelson Island (South Shetland Islands). Close to a refuge, the litter (mainly wood chips, rubber, glass, charcoal, metal and paint fragments sized > 5 cm) had an average density of 33 items m$^{-2}$. In the coastal environment, litter was dominated by plastic debris, mainly originating from fishing and shipping activities, and some anthropogenic debris was also incorporated into seabird nests. Perfetti-Bolaño et al. [50] investigated the occurrence of MPs on surface soils and intertidal sediments along the shores of Fildes Bay (ASPA No. 125; King George Island), an area with six permanent Antarctic stations and an airport that is the major logistical hub for the Antarctic Peninsula. They found the greatest amount of MPs in the soil (mainly fragments 20–500 μm in length, with an average concentration of 0.272 items mL$^{-1}$ of the sample), whereas intertidal sediments were dominated by fibers (length 500–2000 μm; 0.03 items mL$^{-1}$ of the sample). One plastic fiber was also detected in a sediment sample from Ardley Island, an ASPA with no stable human settlements and with only two sporadically used shelters.

Although MP particulates may represent a more likely target for ingestion by aquatic organisms, the recent discovery of MPs in the gut of the Antarctic collembolan *Cryptopygus antarcticus* from King George Island [51], reveals a potential direct impact of plastic debris also in terrestrial ecosystems. Biotic communities in Antarctic ice-free areas are dominated by microbial and cryptogam communities with few species of invertebrates such as nematodes, mites and collembolans. Due to the complexity of the soil medium and the difficulties in extracting plastic debris, it is difficult to assess the environmental fate and the potential impact of macro- and MPs in terrestrial ecosystems. However, some preliminary studies show that plastic debris can represent a further anthropogenic physico-chemical stressor for soil communities and food webs. McKay et al. [122], for example, found that plastic fragments in soils support a distinct microbial habitat, which may alter the ecological

functions critical for agricultural soil. Two soil bacteria isolated from samples collected in Victoria Land (East Antarctica) were able to grow on polypropylene microplastics; after 40 days, they had degraded up to 17.3% of their substrate and had caused significant changes in polypropylene functional groups [59].

## 5. Concluding Remarks and Perspectives

The Southern Ocean and Antarctica received the first human visits by sealers and explorers in the early 19th century and the impact of land-based activities only began to expand during the second half of the 20th century. The number of people visiting the region is relatively low and most of the sea, coastal and inland environments remain nearly untouched. Even on the basis of a very limited amount of available data, often obtained with different methodologies, it appears that most of the Southern Ocean and Antarctica are still scarcely affected by plastic contamination. Only on a local scale are human activities significant sources of plastic debris that could potentially interfere with marine and terrestrial organisms. In general, the occurrence and distribution of MPs in Antarctic waters, sediments and marine organisms appear to be consistent with the findings of Onink et al. [123]. These authors analyzed beaching and resuspension scenarios of plastic debris through a Lagrangian particle transport model and found that throughout the first 5 years after entering the ocean, at least 77% of positively buoyant marine plastics were either beached or floating in coastal waters, with limited transport to the open sea. Although direct measurements are lacking, some preliminary studies seem to indicate that the long-range atmospheric transport may be a significant MPs source for Antarctic marine and terrestrial ecosystems. Due to the atmospheric transport of fibers and the fragmentation of marine plastic debris on their route southward, it is likely that there is a comparatively greater occurrence of nanoplastics in Antarctica than in other regions; however, the current methods for extracting and detecting these particles in environmental matrices appear inadequate.

For a better knowledge of the environmental fate of MPs in Antarctica, it is urgent to investigate further the potential role of snow and sea ice in the entrapment of MPs and in favoring the uptake of these contaminants by ice algae and the Antarctic krill, the key organisms in the Antarctic marine food web. Some preliminary ecotoxicological studies indicate that *E. superba* has the ability to expel the ingested plastic particles; however, the finding that ingested particles can be fragmented into much smaller pieces [113] raises the problem of assessing their environmental fate. Furthermore, considering the very narrow tolerance of Antarctic marine organisms to change in environmental conditions, future MP exposure experiments should consider possible additional stressors, such as warming and acidification of waters and, in the most impacted coastal ecosystems, the presence of other anthropogenic contaminants.

In writing this review, the greatest difficulty was to make reliable comparisons between the available literature data; moreover, the bulk of knowledge has been obtained only in two regions most impacted by human activities (the Antarctic Peninsula and the Ross Sea) and for now, it seems impossible to assess the occurrence of MPs on a continental or Southern Ocean scale. To achieve this goal, it will be necessary to fill methodological gaps in the isolation and characterization of MPs in all environmental matrices. As discussed recently, for instance by Vighi et al. [124], there is an urgent need to harmonize and standardize sampling procedures, and to develop more accurate and possibly less expensive and time-consuming analytical methods. Furthermore, it is necessary to acquire a better knowledge of the degradation processes of plastic in the particular environmental conditions of the Southern Ocean.

Some investigations carried out in Antarctica underline the occurrence in the waters of the Southern Ocean of paint fragments coming from vessels, which can have biological effects through the direct ingestion or adsorption of leached contaminants. Moreover, plastic and paint fragments can accumulate biofouling, and by sinking they can also affect benthic communities. Plastic and paint debris carrying microorganisms and invertebrates

can be perceived as food by marine consumers; therefore, it is necessary to study the composition of the epi-plastic and epi-paint communities, their possible ingestion and the transfer of chemicals and associated microorganisms along trophic chains. With the change in climate and environmental conditions, one major risk, especially for the ecosystems of the sub-Antarctic islands and the Antarctic Peninsula, is the introduction of invasive and pre-adapted species [125]. Available data suggest that propagules of alien species can reach Antarctica not only with scientists, tourists, fishermen or migratory birds but also through long-range transport of plastic debris with air and water masses.

In spite of the Madrid Protocol entered into force 25 years ago and providing principles for environmental protection south of 60° S latitude, there is an increasing number of reports on the occurrence of anthropogenic waste and macro- and microplastic contamination within some ASPAs. This fact strongly suggests the adoption of additional regulation and integrated monitoring and management approaches to address MPs and other specific impacts in the context of local environmental conditions and climate change [126]. Due to the documented impact of plastic fibers and other contaminants of emerging interest [90], it is advisable to equip scientific stations and vessels with suitable sewage treatment plants whenever possible. National Programs for Scientific Research in Antarctica and tour operators should ensure that all people are aware of the potential introduction of alien species, plastics and other persistent environmental contaminants. While MPs, persistent organic pollutants, metals and artificial radionuclides can affect all Antarctic environments through long-range atmospheric and marine transport, the human presence is directly impacting an increasing number of previously pristine and close to pristine areas. The Protocol (Annex V, Art. 3, 2a) states that parties shall seek to identify ASPAs, including "areas kept inviolate from human interference so that future comparisons may be possible with localities that have been affected by human activities". Probably, this would be a good opportunity to introduce further spatial protection. To this end, it would seem appropriate to promote research projects to identify habitats of high priority for conservation and those with a high risk of impact from visiting researchers and tourists, including possible biological invasion, such as areas with geothermal resources and/or affected by rapid climatic and environmental changes.

Most studies on plastic debris in Antarctica have pointed out that the problem is poorly understood and concluded that more data will be needed to assess its impact and develop adequate management policies and regulations. Nonetheless, even though MPs seem to date to have limited or undetectable direct biological effects, the available data show that anthropogenic wastes and plastic debris are affecting even some ASPAs. Therefore, it is necessary that all parties to the Antarctic Treaty and all persons involved in scientific, tourism or fishing activities in Antarctica develop greater awareness of the problem and more responsible behaviors. In addition to promoting measures to monitor and mitigate the risk of plastic pollution in Antarctica, future research should go deeper into the local terrestrial plastisphere, and in particular plastic colonization, trophic transfer and toxicity in soil microorganisms and microfauna, to reveal the real implications for the structure and functions of the fragile Antarctic soil communities.

**Author Contributions:** Conceptualization, R.B.; investigation and structure, R.B. and E.R.; data curation, R.B., E.R., E.B. and I.C.; writing—original draft preparation, R.B. and E.R.; writing—review and editing, E.R., E.B., I.C. and R.B.; visualization, R.B., E.R. and E.B.; supervision, R.B. All authors have read and agreed to the published version of the manuscript.

**Funding:** This research received no external funding.

**Institutional Review Board Statement:** Not applicable.

**Informed Consent Statement:** Not applicable.

**Data Availability Statement:** Not applicable.

**Conflicts of Interest:** The authors declare no conflict of interest.

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
