# Peer review of "Macro- and Microplastics in the Antarctic Environment: Ongoing Assessment and Perspectives"

_environments, doi:10.3390/environments9070093_

Round 1

Reviewer 1 Report

Dear Authors,

I found this article interesting and provides  information for understanding MP pollution caused by global climate change and human activities. However it does not contain photo graphs or diagrams that would greatly facilitate the reader's understanding of this topic.
Moreover, the article requires corrections, in my opinion. Some specific comments are listed below.
1.     Maybe it is worth introducing charts and pictures in the manuscript not just plain text
2.  This is a scientific journal, therefore, the text should be enriched and the literature expanded.
I appreciate the manuscript presented, but it should be modified.

Reviewer 2 Report

Review: Macro- and Microplastics in the Antarctic Environment: Ongo-ing Assessment and Perspectives

The manuscript discusses what is known about the occurrence of plastics in the Antarctic, both macro and microplastics. Overall, it is well written and apart from a number of textual glitches I have very little critical remarks. See the details below.

Introduction:

Line 59: “make up one the most …” -> “make up one of the most …”

Line 85: “Thus, despite …” -> “Despite …” – the word “thus” doesn’t fit with the first threat that is mentioned (it was not referred to before).

Line 101: “a flourishing” – that is a word that bears the wrong connotations. There is nothing bright and sunny about the fact that such studies are necessary.

Line 124: “with a biota” -> “with biota”

Line 131: “Therefore, this review” -> “This review” – there is no causal relationship.

Sources and occurrence:

Lines 178-180: One would expect to see numbers here …

Line 196: “0.01 +/- 0.01 …” – these concentrations are very low! Can one still speak of contamination? Even the concentration of paint fragments would be very low.

Line 206: “originating from the ship” – does that mean that the researchers’ ship was responsible for these paint chips?

Line 207: “Sauria” – listed in the references as “Suaria”. Which is the correct one?

Lines 218-223: It is not entirely clear which type of particles were studied in the two studies cited.

Lines 244 and 250: Make sure the number and the unit remain on the same line.

Lines 261-263: Why is the settling behaviour of the MPs different from other particulate matter? It might be density, but why would these particles, clearly heavier than air, not settle down in the atmosphere?

Lines 268-269: “can act a vector …” – that effect is very limited. There cannot be more mass for the pollutants than the mass of the plastic (as the measured mass will include the pollutant) and the mass of the plastic particles is very small indeed.

Line 284: “King George Is.” – the abbreviation of “Island” to “Is.” Is superfluous (it saves you three characters out of a word of six) and it is annoying because it disturbs the word image. Besides the abbreviation is not used consistently – see line 302. Please write it in full throughout the manuscript.

Line 398: “concentrations” -> “concentration”

Line 405: “respectively” – what two global emissions are meant here? I could not figure it out.

Lines 408-410: These remarks call for a quantification. Given that the amount of microplastics is still very small, how much absorption or scattering are we talking about?

Line 426: “average concentrations” -> “average concentration”

Line 426: “of 13.6 items 50 ml^-1” – very odd-looking unit. I understand that in the original article samples of 50 ml were used, but certainly as written down it looks bizarre. I suggest to convert it to items per litre. Also on line 427.

Conclusions:

Line 446: “Conclusive” -> “Concluding”

Line 465: “relatively greater” -> “greater” – “greater” is always relative, so no need to stress that.

Line 508: “increasing reports” – the reports themselves do not increase, it is their number.

Reviewer 3 Report

This is a valuable review. The authors systematically present the current situation under the topic. I think it can be published after minor revisions:

1.The authors should make recommendations for the future research direction.

2.Using figures and tables works better than text in some places

Round 2

Reviewer 1 Report

Dear Authors,

This article in this version is interesting and provides  information for understanding MP pollution caused by global climate change and human activities. I thank you very much for introducing the graphics and table, it enriched the research presentation and correction references.
The manuscript is well organized, now. The problem statements agree with the title and have significance. The methods used to gather the data for this article were clearly explained.  The quality of citations is good (because in this moment publish a lot of new articles), autors have referenced the interesting works in this field of research. The topic is interested and the result are concreted and useful for the scientific community.

I acceptance and appreciate the manuscript presented.